# Development of a Real-Time Quantitative PCR Assay for the Specific Detection of *Bacillus velezensis* and Its Application in the Study of Colonization Ability

**DOI:** 10.3390/microorganisms10061216

**Published:** 2022-06-14

**Authors:** Shuai Xu, Xuewen Xie, Yanxia Shi, Ali Chai, Baoju Li, Lei Li

**Affiliations:** Institute of Vegetables and Flowers, Chinese Academy of Agricultural Sciences, Beijing 100081, China; xsh1122a@163.com (S.X.); xiexuewen@caas.cn (X.X.); shiyanxia813@163.com (Y.S.); chaiali@163.com (A.C.)

**Keywords:** *Bacillus velezensis*, specific primer, real-time qPCR, rapid detection, colonization ability

## Abstract

*Bacillus velezensis* is a widely used biocontrol agent closely related to *B. amyloliquefaciens*, and the two species cannot be distinguished by universal primers that are currently available. The study aimed to establish a rapid, specific detection approach for *B. velezensis*. Many unique gene sequences of *B. velezensis* were selected through whole genome sequence alignment of *B. velezensis* strains and were used to design a series of forward and reverse primers, which were then screened by PCR and qPCR using different *Bacillus* samples as templates. The colonization ability of *B. velezensis* ZF2 in different soils and different soil environmental conditions was measured by qPCR and a 10-fold dilution plating assay. A specific primer pair targeting the sequence of the D3N19_RS13500 gene of *B. velezensis* ZF2 was screened and could successfully distinguish *B. velezensis* from *B. amyloliquefaciens*. A rapid specific real-time qPCR detection system for *B. velezensis* was established. *B. velezensis* ZF2 had a very strong colonization ability in desert soil, and the optimal soil pH was 7–8. Moreover, the colonization ability of strain ZF2 was significantly enhanced when organic matter from different nitrogen sources was added to the substrate. This study will provide assistance for rapid specificity detection and biocontrol application of *B. velezensis* strains.

## 1. Introduction

*Bacillus velezensis* is a Gram-positive, rod-shaped, motile, spore-forming, aerobic bacterium that has been used as a biocontrol agent for many plant diseases [1]. *B. velezensis* was described in 2005 and first reported as a heterotypic synonym of *B. amyloliquefaciens* based on DNA–DNA relatedness values [2]. Later, *B. velezensis* and *B. amyloliquefaciens* were distinguished based on core genome sequences [3] or phylogenetic analysis of polygenes; however, the two species cannot be distinguished by universal primers that are currently available. Previous studies showed that the specific genes *trpE*, *yecA* and *tetB* were used to specifically detect three closely related *B. amyloliquefaciens* strains, UCMB5033, UCMB5036 and UCMB5113, respectively [4]. The primer pair designed based on the unique gene *ukfpg* of *B. amyloliquefaciens* TF28 was able to distinguish strain TF28 from other *Bacillus* strains [5]. In addition, a colorimetric assay easily distinguished *B. subtilis* from *Escherichia coli* and *Staphylococcus aureus* [6]. However, very few studies have been used to specifically detect *B. velezensis* strains or distinguish *B. velezensis* from *B. amyloliquefaciens*. Therefore, a rapid and specific identification method for *B. velezensis* is necessary for the detection of specific strains in the environment.

*B. velezensis* is widely distributed in various environments, and an increasing number of *B. velezensis* strains have been used as biocontrol agents for plant disease. For example, *B. velezensis* FZB42^T^ was reported to produce many kinds of lipopeptides and showed direct suppression of many plant pathogens [7]. *B. velezensis* SQR9 could stimulate resident rhizosphere-beneficial microorganisms and protect plants against diseases [8]. *B. velezensis* QST713 showed broad-spectrum antimicrobial or antibacterial activities and has been used as an antagonist against green mold disease [9]. *B. velezensis* ZF2 has been reported as a potential biocontrol agent that shows a broad spectrum of antagonistic activities against many plant pathogens and exhibits strong inhibitory activity against *Corynespora* leaf spot disease in cucumber [10]. However, taking plant-beneficial microorganisms from lab to agricultural application is a great challenge, as exogenous inoculum is usually eliminated in soils due to competition from indigenous microbes or complicated soil environments [11,12]. Research on these biocontrol agents indicated that the biocontrol efficacy of the microbes to control plant diseases was related to their ability to survive and maintain an abundant population in rhizosphere soil [13]. Therefore, it is important to detect the viability of inoculum in soil. The current detection method is a 10-fold dilution plating test, which is time-consuming and laborious. Therefore, a rapid detection system for *B. velezensis* is essential for determining the colonization ability of strains in biocontrol applications.

It is worth noting that many *B. velezensis* strains do not show excellent control effects for plant disease in field applications, although these strains show broad-spectrum antagonistic activities against phytopathogens. One possible explanation is that antagonistic bacteria cannot rapidly colonize the rhizosphere and soil because of the complicated environmental conditions [14]. It is important to explore the colonization ability of *B. velezensis* under different environmental conditions for rational biocontrol application of strains.

In this study, we screened a pair of primers that could distinguish *B. velezensis* from *B. amyloliquefaciens*. A new, rapid and specific detection system was established that facilitates rapid detection of the *B. velezensis* genus. Moreover, we measured the colonization ability of *B. velezensis* ZF2 in different soils and different soil environmental conditions, including pH and nutrient elements. Our study will promote the application and detection of *B. velezensis* in biocontrol applications.

## 2. Materials and Methods

### 2.1. Strains, Culture Conditions and DNA Extraction

The bacterial reference strains used in this study are listed in Table 1. All of the test strains used in the study were cultivated in Luria broth medium (LB) or nutrient broth (NB) at 28 °C with shaking for 24 h. Genomic DNA was extracted from the cultured cells (OD600 = 0.8) using a TIANamp Bacteria DNA kit (Tiangen Biotech (Beijing) Co., Ltd., Beijing, China).

### 2.2. Design and Selection of Species-Specific Primers

The sequences of the housekeeping genes *gyrB*, *gap*, *rpoD*, *atpD*, *rho*, 16S rRNA and other functional genes *galE*, *metC*, *pdhA*, and *pgk* of strains ZF2, FZB42, DSM 7, and 168 were obtained from the corresponding whole genome sequence (GenBank: CP032154.1, CP000560.1, FN597644.1, AL009126.3, respectively). Every gene sequence of *B. velezensis* ZF2 and *B. velezensis* FZB42 was aligned with those of the *Bacillus* strains DSM 7 and 168 using DNAMAN 7.0 [15]. Sequence regions unique to *B. velezensis* were used to design a series of forward and reverse primers. Our previous study showed that the sequence of the *fliC* gene (coding flagellin) from strains ZF2 and FZB42 exhibited low homology to strains DSM 7 and 168 (88% and 56%, respectively) [10], so the unique region sequence of the flic gene in *B. velezensis* ZF2 was also used to design a series of forward and reverse primers based on the flic sequence alignment results. In addition, to increase the possibility of primer specificity for the *B. velezensis* species, the whole genome sequence of *B. velezensis* ZF2 was compared with other *B. velezensis* strains and used to design a series of forward and reverse primers by Primer Premier 5 [16]. As a result, 26 pairs of primers were designed (Appendix A) and synthesized by Biomed Biotech (Beijing, China) Co., Ltd. and were screened by PCR using diverse DNA templates of different *Bacillus* strains.

### 2.3. Primer Specificity Verification and Real-Time qPCR Assays

All primers were screened by PCR using different bacterial genomic DNA samples, including *B. velezensis* ZF2, *B. velezensis* ZF128, *B. velezensis* FZB42^T^ [7], *B. amyloliquefaciens* DSM 7^T^ [17] and *B. subtilis* ZF168^T^ [18]. The primer with the best specificity for *B. velezensis* was tested against diverse bacterial genomic DNA templates, including *B. velezensis* ZF2, *B. velezensis* ZF128, *B. velezensis* ZF145, *B. velezensis* LS69 [19], *B. velezensis* SQR9 [8], *B. velezensis* FZB42^T^, *B. amyloliquefaciens* 75, *B. amyloliquefaciens* DSM 7^T^, *B. subtilis* ZF161, *B. subtilis* 168^T^, *B. safensis* ZF438, *P. polymyxa* ZF129 [20], *P. peoriae* ZF390, *Rahnella aceris* ZF458, *R. aquatilis* ZF7 [21], *Lysobacter enzymogenes* CX03, *Pectobacterium brasiliense* [22], *Pseudomonas amygdali pv. lachrymans* [23] and *Xanthomonas campestris* pv. *campestris* 8004 [24] (Table 1).

Selecting the best primer pair, the specific PCR conditions were optimized, including template DNA (10 ng of genomic DNA) and the annealing temperature (52.0–57.0 °C with approximately 1.0 °C increments). Amplifications were performed in a 20 μL reaction mixture including 10 μL of 2 × Taq DNA polymerase mix (Biomed Biotech (Beijing, China) Co., Ltd.), 1 μL of the genomic DNA template, 1 μL of each primer (10 μM) and 7 μL sterile water. Negative controls were included for each PCR assay. The PCR procedure consisted of one cycle of 3 min at 94 °C and 30 cycles of 30 s at 94 °C, 30 s at 58 °C, and 30 s at 72 °C, and a final extension step was run for 5 min at 72 °C. After amplification, 10 μL of each PCR product was analyzed through electrophoresis in a 1% agarose gel in 1× Tris-acetate-EDTA (TAE) buffer, stained with ethidium bromide and visualized using an ultraviolet transilluminator.

After PCR validation, the screened primers were verified by qPCR. All qPCR assays were performed on an Applied Biosystems 7500 Real-Time PCR System (Applied Biosystems, Waltham, MA, USA) [25] with a MicroAmp^®^ Optical 96-Well Reaction Plate closed with the MicroAmp^®^ Optical 8-Cap Strip (Applied Biosystems). The reaction was performed in a final volume of 20 μL containing 1 μL of the genomic DNA template, 10 μL of 2 × SYBR^®^Green PCR Mastermix (TIANamp), 0.4 μL ROX, 0.5 μL of each primer (10 μM) and 7.6 μL sterile water. The following thermal program was applied: a single cycle of DNA polymerase activation for 10 min at 98 °C followed by 40 amplification cycles of 15 s at 98 °C (denaturing step) and 32 s at 60 °C (annealing and extension step). The melting curve and CT value were used to assess the specific amplification and amplification efficiency.

### 2.4. Standard Curve Determination and Sensitivity Test

After determining the optimal specific primer, the corresponding purification amplification fragment was cloned into the pMD18-T vector by heat shock transformation and copied into *Escherichia coli* DH5α according to a previously reported method [26]. Plasmid DNA containing the target gene was extracted from cultured DH5α cells using a TIANamp plasmid DNA kit (Tiangen Biotech (Beijing, China) Co., Ltd.). Standard curves were generated using 10-fold serial dilutions (concentration from 10^8^ fg·μL^−1^ to 1 fg·μL^−1^) of the plasmid containing the fragment copy of the optimal targeted gene as described by a previous study [27]. The abundance of *B. velezensis* was determined using SYBR Green assays with the selected primers. Each assay was performed in triplicate, and a linear relationship equation was established between the plasmid DNA concentration and the CT value. The copy number was calculated with the equation N = CM, where N is the sample copy number, C is the concentration of DNA, and M is the mean mass of the genomic DNA [28]. The minimum detection limit was calculated according to the linear relationship equation.

### 2.5. Colonization Ability of B. velezensis ZF2 in Different Soils

*B. velezensis* ZF2 was marked as rifampicin resistant according to a previously reported method with modifications [29]. Rifampicin-resistant mutants of strain ZF2 were obtained by transferring colonies to LB medium containing increasing concentrations (1, 5, 10, 20, 30, 40, 50 ng·μL^−1^) of rifampicin. After that, equal suspensions of rifampicin-resistant strain ZF2^rif+^ (OD_600_ = 1.0) were mixed in different soils at equivalent weights (including black soil, red soil, yellow brown soil, brown soil and desert soil), and the colonization ability of the strain in different soils was determined using a standard 10-fold dilution plating assay as previously described [30]. For quantification of the ZF2 density, three aliquots (100 mL) per dilution were spread on LA agar medium with rifampicin (50 ng·μL^−1^), and the plates were incubated at 28 °C for 2 days prior to colony counting. The soil samples were detected weekly after mixing with the strain ZF2 suspension (1, 7, 14, 21, 28, and 35 days), and each sample was tested three times. In addition, the genomic DNA of diverse soil samples in different periods was extracted using a TIANamp Bacteria DNA kit (Tiangen Biotech (Beijing, China) Co., Ltd.), and the corresponding natural soils were used as negative control. Then, the colonization ability of ZF2 in different soils was detected using the rapid specific detection system established above.

### 2.6. Determination of Physicochemical Properties of the Different Soils

The physicochemical properties, including pH, organic matter, total N, ammonium N, nitrate N, total P, available P, total K and available K, of the five kinds of soils were measured at the China National Rice Research Institute.

### 2.7. Colonization Ability of B. velezensis ZF2 under Different Environmental Conditions (pH and Nutrient Elements)

To understand the effect of different environmental conditions on the colonization ability of *B. velezensis* ZF2^rif+^, the nursery substrate with a mixed-strain ZF2 suspension (OD_600_ = 1.0) was placed in plastic boxes and incubated at different pH values (4, 5, 6, 7, 8, 9 and 10). The boxes were placed in an incubator to maintain stable environmental conditions. The colonization ability of strain ZF2^rif+^ in different substrate environmental conditions was detected using the rapid, specific real-time qPCR detection system and the standard 10-fold dilution plating assay every week, and each sample was tested three times.

Similarly, the effect of different nutrient elements on the colonization ability of *B. velezensis* ZF2^rif+^ was tested in the same way. Corn flour, yeast extract, peanut flour, peptone, soybean flour and soluble starch were selected as nitrogen sources; maltose, sucrose, fructose, dextrin, and molasses were selected as carbon sources; MgCl_2_, CaCl_2_, FeSO_4_, KCl and NaCl were selected as the inorganic compounds. The colonization ability of strain ZF2^rif+^ in substrate samples mixed with different nutrient elements was detected using the rapid, specific real-time qPCR detection system and the standard 10-fold dilution plating assay every week, and each sample was tested three times.

## 3. Results

### 3.1. Establishment of Specific Detection System

#### 3.1.1. Design and Selection of Specific Primers

Through multiple sequence alignment, the sequence identities of the four genes 16S rRNA, *gap*, *rpoD* and *rho* between ZF2 and other *B. amyloliquefaciens* or *B. subtilis* strains were over 98%, while the sequence identities of the genes *galE*, *gyrB*, *pdhA*, *pgk*, *fliC* and *metC* among the four strains were under 95%. Primers targeting the unique region sequences of the *galE*, *gyrB*, *pdhA*, *pgk*, *fliC* and *metC* genes from *B. velezensis* ZF2 were designed. Another 12 pairs of primers were designed based on the unique gene sequence of *B. velezensis* (Appendix A). After PCR screening, no primers targeting the housekeeping genes or flic gene could distinguish *B. velezensis* from other *Bacillus* species. As expected, primer pairs based on the unique gene sequence of *B. velezensis* were evaluated for their ability to distinguish *B. velezensis* from the closely related species *B. amyloliquefaciens* and *B. subtilis*. One pair of primers targeting the sequence of gene D3N19_RS13500 (Figure 1) yielded amplification products of the predicted size (192 bp) only in the *B. velezensis* template except for the strain FZB42^T^ but not in the *B. amyloliquefaciens* and *B. subtilis* templates (Figure 2). The targeting gene could be retrieved in many *B. velezensis* strains in the NCBI database, and the sequence identities were over 94% (Table 2). Therefore, the pair of primers amplifying the D3N19_RS13500 gene fragments was selected for subsequent verification.

#### 3.1.2. Verification of Primer Specificity and Optimization of PCR Reaction Conditions

To verify the specificity of the D3N19_RS13500 primer, diverse bacterial genomic DNA was used in PCR and real-time qPCR detection. The results showed that only the primer targeting D3N19_RS13500 gave amplification products of the predicted size (192 bp) in *B. velezensis* strains except for FZB42^T^ and SQR9, but not in other bacterial strains (Figure 3A). To improve the amplification efficiency, the annealing temperature was optimized by performing gradient PCR. A unique clear target product was obtained at annealing temperatures ranging from 54–57 °C (Figure 3B). As expected, a fluorescence signal appeared when using the genomic DNA of *B. velezensis* as the template in the real-time qPCR (Figure 3C).

#### 3.1.3. Standard Curve Determination and Detection Limit

Different amplification curves were generated using different concentrations of genomic DNA as templates (Figure 4A). Standard curves of real-time PCR displayed dynamic ranges on 8 log DNA dilutions (Figure 4B). All of the qPCR assays were performed in a linear manner with the linear equation y = −3.875x + 30.713 (y represents the CT value, x represents the lg DNA concentration), and the R^2^ value was 0.9972. The results from the dynamic range analyses allowed the determination of PCR efficiency, as the E value was 81%. The R^2^ and E values of the developed SYBR^®^ Green qPCR complied with the acceptance limits. According to the linear equation, the minimum detection concentration and the minimum detection gene copies were calculated as 0.1 fg·μL^−1^ and 474 copies·μL^−1^, respectively.

### 3.2. Detection of the Colonization Ability of B. velezensis ZF2

#### 3.2.1. Colonization Ability of *B. velezensis* ZF2 in Different Soils

The colonization ability of strain ZF2 in different soils was measured by a real-time qPCR rapid detection system. The results showed that strain ZF2 had a maximum gene copy number in desert soil (approximately 10^8^ copy numbers in 1 g soil) (Table 3). To verify the results of qPCR detection, the colonization ability of *B. velezensis* ZF2 in diverse soils was tested by a 10-fold dilution plating assay. As expected, the maximum number of ZF2 colonies was isolated from the desert soil (over 10^8^ cfu·g^−1^ in 35 days) (Table 3), followed by black soil, yellow brown soil, brown soil and red soil. These results revealed that the ZF2 strain exhibited the strongest colonization ability in desert soil (from Ningxia Province), which was consistent with the qPCR detection results.

#### 3.2.2. Determination of Soil Physicochemical Properties

To understand the different colonization abilities of ZF2 in different soils, the physicochemical properties of diverse soils were measured. The desert soil had a higher pH value (8.40) than other types of soils and had higher nitrate N (43.75 mg·kg^−1^) and ammonium N (24.02 mg·kg^−1^) than other types of soils (Table 4). These data indicated that the colonization abilities of strain ZF2 in different soils may be associated with the soil physicochemical properties, especially the pH, ammonium N and nitrate N.

#### 3.2.3. Colonization Ability of *B. velezensis* ZF2 under Different Environmental Conditions (pH)

The results showed that soil pH was associated with the colonization ability of strain ZF2. Strain ZF2 showed the strongest colonization ability when the soil pH value was 7 to 8 (the number of colonies and gene copies were approximately equal to 10^7^·g^−1^ substrate), while the number of ZF2 colonies and gene copies were less than 10^7^·g^−1^ substrate when the pH value was over 8 or below 7 (Table 5). These results indicated that neutral or weakly alkaline soil conditions may be favorable for the colonization of *B. velezensis* ZF2.

#### 3.2.4. Colonization Ability of *B. velezensis* ZF2 under Different Nutrient Additions (Carbon Source, Nitrogen Source, Inorganic Compounds)

Furthermore, the effect of different nutrient additions on the colonization ability of strain ZF2 in the nursery substrate was measured in the same way. The results indicated that adding different carbon sources or inorganic compounds did not enhance the colonization ability of the ZF2 strain, as the number of ZF2 colonies remained the same as that in the control or less than that in the control. Interestingly, the number of ZF2 colonies and gene copies were 10^5^−10^6^ when different nitrogen sources were added to the nursery substrate, while the number of colonies and gene copies were only 10^4^ in the control nursery substrate at 35 days (Table 6). These results indicated that adding a nitrogen source to the soil significantly enhanced the colonization ability of strain ZF2.

## 4. Discussion

*Bacillus velezensis* is widely distributed in soil, water, and plants, and has been used for controlling plant disease due to its direct or indirect growth improvement effect on many plants [31,32]. *B. velezensis* ZF2, isolated from the stem of cucumber, has been reported to have broad-spectrum antagonistic activities and a significant ability to control *Corynespora* leaf spot disease [10]. In this study, a rapid, specific real-time qPCR detection system for *B. velezensis* based on the unique gene sequence of strain ZF2 was established, and its application in measuring the colonization ability of strain ZF2 was studied.

Housekeeping genes, including 16S rRNA, *gyrB*, *gap*, and *rpoD*, are widely used for the identification and classification of bacteria [33], and real-time quantitative polymerase chain reaction is often used for the rapid detection of bacteria [34]. However, a single gene usually cannot distinguish a species from other closely related species. A primer pair targeting the 16S rRNA gene of *B. pumilus* failed to distinguish *B. megaterium*, *B. circulans* and *Paenibacillus mucilaginosus*. It was reported that the *gyrB* gene has a relatively higher discrimination ability than the 16S rRNA gene, and it was used for the specific identification of *P. mucilaginosus* [35]. However, a primer pair targeting the *gyrB* gene of *B. velezensis* ZF2 failed to distinguish *B. velezensis* from *B. amyloliquefaciens* and *B. subtilis*. Moreover, primer pairs targeting other housekeeping genes could not distinguish *B. velezensis* from *B. amyloliquefaciens* and *B. subtilis* due to the close relationships among *B. velezensis*, *B. amyloliquefaciens* and *B. subtilis* [3].

In recent years, many non-conserved genes have been used to identify bacterial species. For instance, the phosphoenolpyruvate/sugar phosphotransferase system I gene and the adenylosuccinate synthetase gene were reported to discriminate *B. anthracis* from *B. cereus* [36]. In this study, primer pairs targeting the *fliC* gene (which showed low homology among *B. velezensis*, *B. amyloliquefaciens* and *B. subtilis*) of strain ZF2 were designed for the rapid identification of *B. velezensis* [10]. However, these primer pairs could not distinguish *B. velezensis* from *B. amyloliquefaciens* and *B. subtilis*. Satisfactorily, a primer pair targeting the gene D3N19_RS13500 of strain ZF2 showed specific amplification only from genomic DNA of *B. velezensis* strains. Interestingly, the 192 bp product was not amplified when using *B. velezensis* FZB42^T^ and SQR9 as templates in the PCR screening test. This phenomenon might be related to the taxonomic position of the two strains, based on the fact that *B. velezensis* FZB42^T^ and SQR9 were originally classified as *B. amyloliquefaciens* [37,38]. In addition, there was substantial genomic diversity in bacteria, even in strains of the same genus, so, many genera were divided into different subgroups. As the largest group of *Pseudomonas*, *Pseudomonas fluorescens* was divided further into eight or nine subgroups [39]. To explain this interesting phenomenon, more in-depth studies are needed, and it would be necessary to design a separate primer to detect these exclusive strains. However, the gene could be retrieved in most *B. velezensis* strains and its sequence had high homology. These results indicated that the selected primer pair had the ability to distinguish most *B. velezensis* strains from *B. amyloliquefaciens* strains and *B. subtilis* strains; however, in special cases, its specificity needs to be further verified. To our knowledge, this is the first report to distinguish *B. velezensis* from *B. amyloliquefaciens* and *B. subtilis* through a single gene. A rapid and specific detection system is essential to detect the colonization ability of *B. velezensis* strains in biocontrol applications.

*Bacillus* was reported to be a biocontrol agent with the best application potential due to its stability, broad-spectrum antagonism and environmental friendliness. However, the prevention and control effect of *Bacillus* for controlling plant disease is affected by many factors. It was reported that soil bacterial diversity could be strongly affected by pH, soil type, latitude, vegetation, moisture, temperature, and nutrient availability [40]. Among these, soil pH was the best predictor of both soil bacterial diversity and richness, whereas soil type strongly influenced soil bacterial composition [41,42]. Although the general patterns underlying variations in biodiversity have been observed, the influence of these factors on the viability of microorganisms remains unclear.

Recently, many beneficial microbes have been used as biological control agents to control plant diseases and were reported as a potential way to decrease the negative effect of chemicals on the environment [43]. Successful colonization of biocontrol agents in the rhizosphere soil is a prerequisite for disease control [44]. Soil is a complex environmental matrix, and the soil physicochemical properties have a great influence on the colonization of biocontrol agents [45]. In this study, the colonization ability of *B. velezensis* ZF2 in different types of soils was measured by real-time qPCR detection and a 10-fold dilution plating assay. The results showed that different types of soils had a strong influence on the colonization ability of strain ZF2. *B. velezensis* ZF2 exhibited the strongest colonization ability in desert soil compared with other soils. However, the colonies and gene copies of strain ZF2 decreased in all test soils over time, which indicated that *B. velezensis* ZF2 was degraded in different soils due to the indigenous soil environment. Indeed, many previous studies have reported that the inocula failed to propagate and decreased significantly after inoculation in exogenous soils [46,47,48]. The reason might be that the inocula were inhibited or eliminated by the indigenous microbes and/or local soil conditions [13]. Interestingly, evaluation of the physicochemical properties showed that desert soil had a higher pH value, higher nitrate N and ammonium N than other soils, although the total N in desert soil was very low. Previous studies showed that soil with a higher pH had an estimated bacterial richness 60% higher than that of more acidic soil [41]. As expected, our research found that *B. velezensis* ZF2 had a strong colonization ability in neutral or weakly alkaline soil conditions, which indicated that pH was closely related to the colonization ability of *Bacillus*. Furthermore, adding a nitrogen source to the soil could significantly enhance the colonization ability of strain ZF2, which revealed that available nitrogen could promote the colonization ability of *Bacillus* in soil.

In addition, a 10-fold dilution plating assay showed that strain ZF2 had a strong colonization ability in red soil at the beginning; however, qPCR detection showed the opposite result: a low gene copy number of strain ZF2 emerged in red soil. The reason for this phenomenon may be that red soil has strong adsorption and is not suitable for the extraction of soil genomic DNA. Colonization experiments were conducted under controlled environmental conditions in growth chambers rather than in the field. The advantages of using controlled environmental conditions are obvious in that they enable the assessment of specific factors, and thus, the conditions would not fluctuate and interact with other factors.

## 5. Conclusions

In this study, a pair of specific primers for *B. velezensis* targeting the gene D3N19_RS13500 of strain ZF2 was designed and could distinguish *B. velezensis* from the closely related *B. amyloliquefaciens* and *B. subtilis* species. Real-time polymerase chain reaction assays for the rapid detection of *B. velezensis* were developed, and the minimum number of detected gene copies was 474 copies·μL^−1^. According to the rapid detection method and 10-fold dilution plating assay, strain ZF2 had the strongest colonization ability in desert soil. In addition, this study showed that neutral or weakly alkaline soil conditions might be suitable for the colonization of *B. velezensis*, and adding a nitrogen source to the soil was proven to enhance the colonization ability of *B. velezensis*. Our study provides convenience for rapid detection and biocontrol applications of *Bacillus* strains in agricultural fields.

## Figures and Tables

**Figure 1 microorganisms-10-01216-f001:**
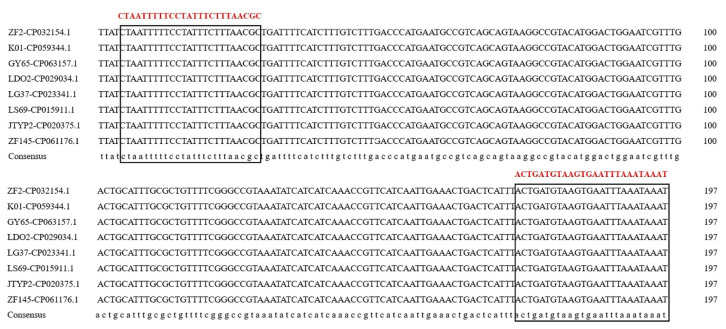
Specific primer of gene D3N19_RS13500 existed in many *B. velezensis* strains. The red fonts indicate the primer sequence. The black fonts indicate the corresponding gene sequence in different *B. velezensis* strains (including ZF2, K01, GY65, LDO2, LG37, LS69, JTYP2 and ZF145).

**Figure 2 microorganisms-10-01216-f002:**
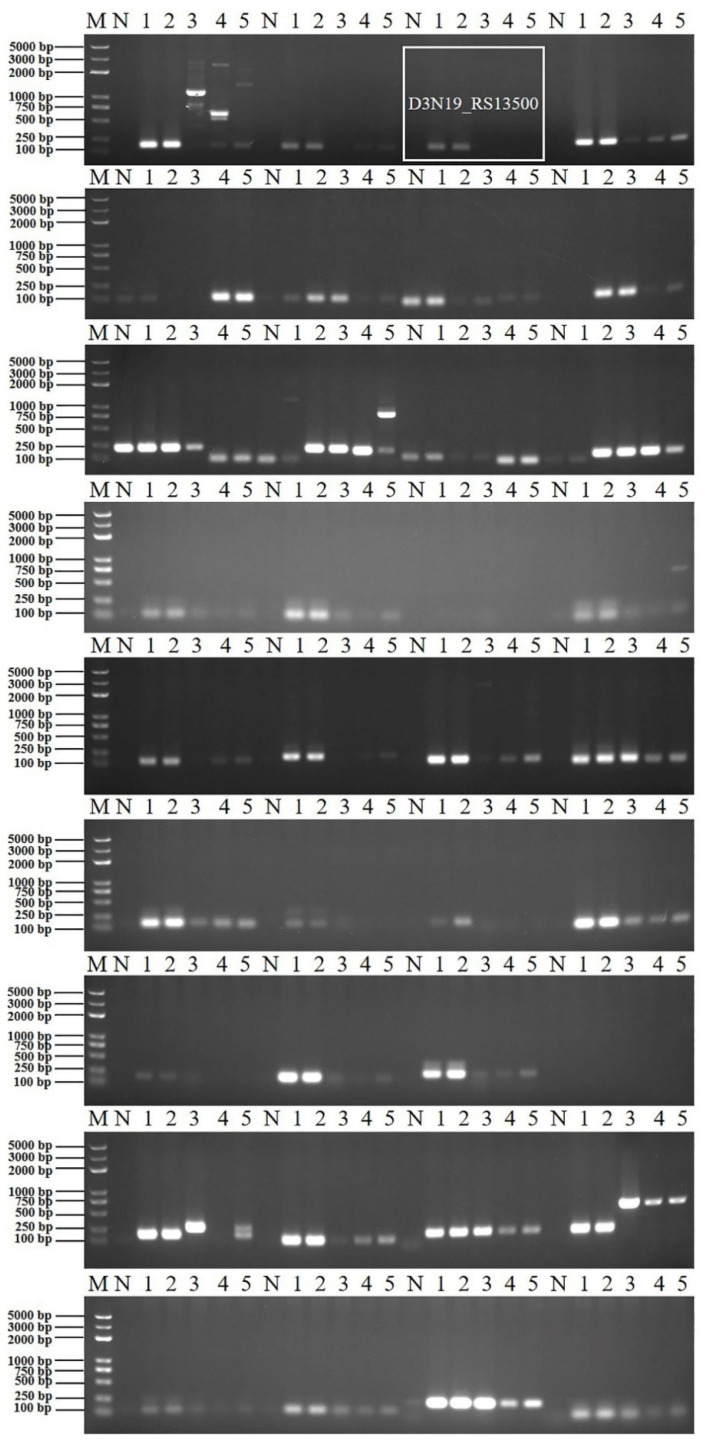
Specific primers targeting *B. velezensis* screening in different *Bacillus* strains. M, DNA marker (from large to small was 5000 bp, 3000 bp, 2000 bp, 1000 bp, 750 bp, 500 bp, 250 bp, 100 bp); N, negative control; 1, amplification template of *B. velezensis* ZF2 genomic DNA; 2, amplification template of *B. velezensis* ZF128 genomic DNA; 3, amplification template of *B. velezensis* FZB42^T^ genomic DNA; 4, amplification template of *B. amyloliquefaciens* DSM 7^T^ genomic DNA; 5, amplification template of *B. subtilis* 161^T^ genomic DNA.

**Figure 3 microorganisms-10-01216-f003:**
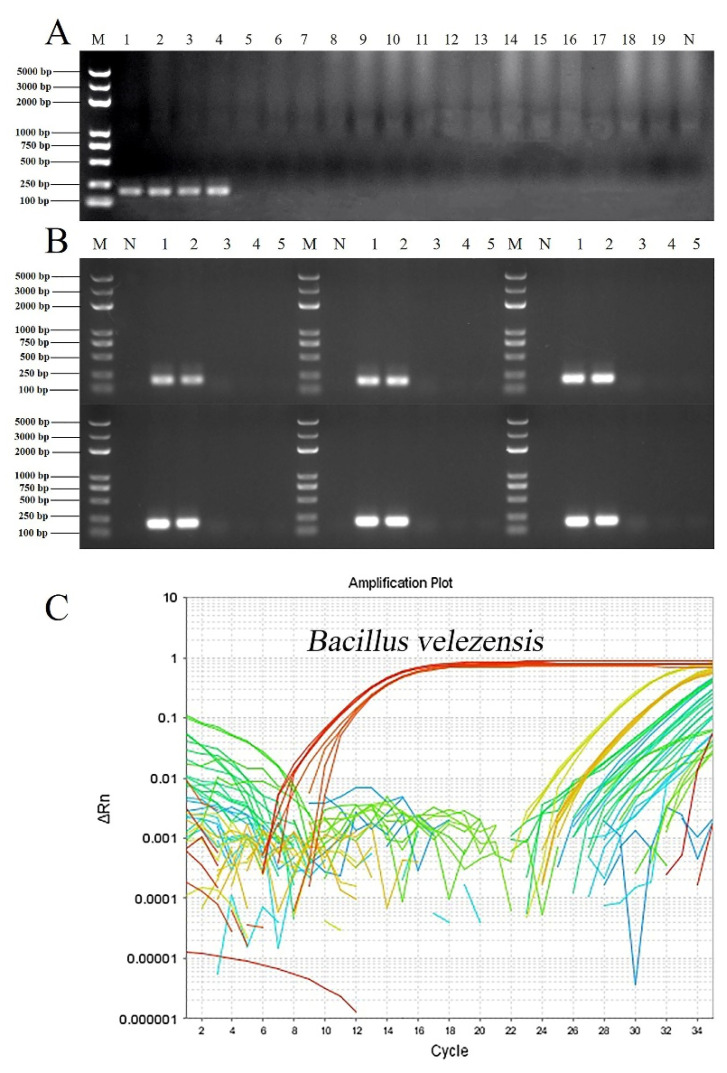
Verification of specific primers targeting *B. velezensis* among different bacterial strains. (**A**), PCR verification. M, DNA marker (from large to small was 5000 bp, 3000 bp, 2000 bp, 1000 bp, 750 bp, 500 bp, 250 bp, 100 bp); N, negative control; 1, amplification template of *B. velezensis* ZF2 genomic DNA; 2, amplification template of *B. velezensis* ZF128 genomic DNA; 3, amplification template of *B. velezensis* ZF145 genomic DNA; 4, amplification template of *B. velezensis* LS69 genomic DNA; 5, amplification template of *B. velezensis* SQR9 genomic DNA; 6, amplification template of *B. velezensis* FZB42^T^ genomic DNA; 7, amplification template of *B. amyloliquefaciens* ZF75 genomic DNA; 8, amplification template of *B. amyloliquefaciens* DSM 7^T^ genomic DNA; 9, amplification template of *B. subtilis* ZF161 genomic DNA; 10, amplification template of *B. subtilis* 168^T^ genomic DNA; 11, amplification template of *B. safensis* ZF438 genomic DNA; 12, amplification template of *P. polymyxa* ZF129 genomic DNA; 13, amplification template of *P. peoriae* ZF390 genomic DNA; 14, amplification template of R. aceris ZF458 genomic DNA; 15, amplification template of *R. aquatilis* ZF7 genomic DNA; 16, amplification template of *L. enzymogenes* CX03, genomic DNA; 17, amplification template of *Pectobacterium brasiliense* genomic DNA; 18, amplification template of *Pseudomonas amygdali* pv. *lachrymans* genomic DNA; 19, amplification template of *Xanthomonas campestris* pv. *campestris* 8004 genomic DNA. (**B**), Screening of the optimal PCR annealing temperature of the specific primer. The temperature gradient was 52 °C, 53 °C, 54 °C, 55 °C, 55 °C, 56 °C, and 57 °C. M, DNA marker (from large to small was 5000 bp, 3000 bp, 2000 bp, 1000 bp, 750 bp, 500 bp, 250 bp, 100 bp); N, negative control; 1, amplification template of *B. velezensis* ZF2 genomic DNA; 2, amplification template of *B. velezensis* ZF128 genomic DNA; 3, amplification template of *B. velezensis* FZB42^T^ genomic DNA; 4, amplification template of *B. amyloliquefaciens* DSM 7^T^ genomic DNA; 5, amplification template of *B. subtilis* 161^T^ genomic DNA. (**C**), Specific primers targeting *B. velezensis* verification among different bacterial strains by qPCR. Curves in different colors represent different amplifications using genomic DNA of different strains as templates.

**Figure 4 microorganisms-10-01216-f004:**
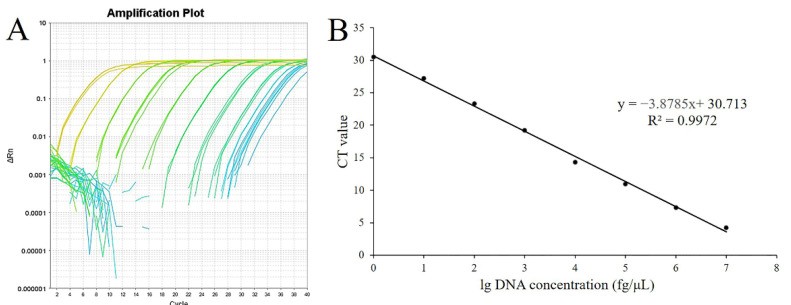
The standard curve for the rapid detection system of *B. velezensis* ZF2. (**A**), Amplification plot of real-time qPCR using different concentrations of plasmid DNA containing the target gene of the *B. velezensis* ZF2 (D3N19_RS13500) fragment. (**B**), Standard curves based on different concentrations of plasmid DNA and the corresponding amplified CT value. X represents the lg DNA concentration, and y represents the CT value.

**Table 1 microorganisms-10-01216-t001:** Reference strains used in this study.

Strains	Source of Strain or Accession
*Bacillus velezensis* ZF2	CP032154.1
*Bacillus velezensis* ZF128	Isolated in lab
*Bacillus velezensis* ZF145	Isolated in lab
*Bacillus velezensis* LS69	CP015911.1
*Bacillus velezensis* SQR9	CP006890.1
*Bacillus velezensis* FZB42^T^	NC_009725.2
*Bacillus amyloliquefaciens* ZF75	Isolated in lab
*Bacillus amyloliquefaciens* DSM 7^T^	NC_014551.1
*Bacillus subtilis* ZF161	Isolated in lab
*Bacillus subtilis* 168	AL009126.3
*Bacillus safensis* ZF438	Isolated in lab
*Paenibacillus polymyxa* ZF129	NZ_CP040829.1
*Paenibacillus peoriae* ZF390	Isolated in lab
*Rahnella aceries* ZF458	Isolated in lab
*Rahnella aquatilis* ZF7	NZ_CP032296.1
*Lysobacter enzymogenes* CX03	Isolated in lab
*Pectobacterium Brasiliense*	CP020350.1
*Pseudomonas amygdali pv. lachrymans*	CP020351.1
*Xanthomonas campestris pv. campestris* 8004	NC_007086.1

**Table 2 microorganisms-10-01216-t002:** Search results for D3N19_RS13500 gene in NCBI database.

Strains	Identities %	Sequence ID	locus_tag
*Bacillus velezensis* ZF2	100.00	CP032154.1	D3N19_13500
*Bacillus velezensis* WLYS23	100.00	CP055160.1	HUW56_06825
*Bacillus velezensis* A2	100.00	CP053717.1	HNV93_RS18185
*Bacillus velezensis* BIM B-1312D	100.00	CP050448.1	HB674_13380
*Bacillus velezensis* UB2017	100.00	CP049741.1	G7X27_RS13380
*Bacillus velezensis* FJAT-52631	100.00	CP045186.1	F9286_13510
*Bacillus velezensis* LC1	100.00	CP044349.1	F6467_13510
*Bacillus velezensis* LG37	100.00	CP023341.1	CMV18_07105
*Bacillus velezensis* ANSB01E	100.00	CP036518.1	EYB46_16460
*Bacillus velezensis* JT3-1	100.00	CP032506.1	D5H27_11180
*Bacillus velezensis* LDO2	100.00	CP029034.1	DDE72_03255
*Bacillus velezensis* DR-08	100.00	CP028437.1	DA376_13590
*Bacillus velezensis* GQJK49	100.00	CP021495.1	BAGQ_RS19690
*Bacillus velezensis* CBMB205	100.00	CP011937.1	AAV34_05955
*Bacillus velezensis* JTYP2	100.00	CP020375.1	BAJT_13390
*Bacillus velezensis* sx01604	100.00	CP018200.1	BLL65_RS19535
*Bacillus velezensis* LS69	100.00	CP015911.1	A8142_RS19645
*Bacillus velezensis* S3-1	100.00	CP016373.1	A5891_RS19675
*Bacillus velezensis* CBMB205	100.00	CP014838.1	BCBMB205_RS13490
*Bacillus velezensis* GUIA	100.00	CP094930.1	MUB47_13415
*Bacillus velezensis* MBI600	100.00	CP094686.1	MTR96_14500
*Bacillus velezensis* AP3	100.00	CP094294.1	MRS49_RS03450
*Bacillus velezensis* JJ47	100.00	CP091288.1	L3C11_13755
*Bacillus velezensis* AB191	100.00	CP089996.1	LVY83_13415
*Bacillus velezensis* B4-7	100.00	CP080760.1	K3A92_13320
*Bacillus velezensis* BS-G1	100.00	CP078149.1	KV103_RS13320
*Bacillus velezensis* J17-4	100.00	CP060420.1	H8P15_13320
*Bacillus velezensis* KOF112	100.00	AP024603.1	KJS48_RS13235
*Bacillus velezensis* WSM-1	100.00	CP068989.1	IPZ53_13320
*Bacillus velezensis* GY65	100.00	CP063157.1	IMZ24_05720
*Bacillus velezensis* BSC16a	100.00	CP062074.1	IGB10_13320
*Bacillus velezensis* ZF145	100.00	CP061176.1	IAQ68_13320
*Bacillus velezensis* YB-130	100.00	CP054562.1	HUF97_RS13715
*Bacillus velezensis* K01	100.00	CP059344.1	HU024_03380
*Bacillus velezensis* ZeaDK315	95.94	CP043809.1	ETZ92_008920
*Bacillus velezensis* K26	95.94	CP023075.1	CK238_14205
*Bacillus velezensis* SGAir0473	95.94	CP027868.1	C1N92_05090
*Bacillus velezensis* 10075	95.94	CP025939.1	C0W57_02140
*Bacillus velezensis* ATR2	95.94	CP018133.1	BMJ37_RS20275
*Bacillus velezensis* SRCM100072	95.94	CP021888.1	S100072_02830
*Bacillus velezensis* C1	95.94	CP064091.1	IRJ21_RS06175
*Bacillus velezensis* K203	95.94	CP092185.1	MF598_05880
*Bacillus velezensis* LOH112	95.94	CP092110.1	LGL65_RS16875
*Bacillus velezensis* Pilsner2-2	95.94	OU015476.1	NA
*Bacillus velezensis* Pilsner1-2	95.94	OU015424.1	NA
*Bacillus velezensis* SWUJ1	95.94	CP077672.1	KTT68_RS13370
*Bacillus velezensis* AD-3	95.94	AP024501.1	BVAD3_RS21170
*Bacillus velezensis* NST6	95.94	CP063687.1	BACVE_RS14395
*Bacillus velezensis* JSRB 08	95.94	CP059497.1	H2N97_14425
*Bacillus velezensis* ONU553	94.92	CP043416.1	FZE25_RS19965
*Bacillus velezensis* At1	94.92	CP041145.1	D073_RS19585
*Bacillus velezensis* AP183	94.92	CP029296.1	RZ52_RS20115
*Bacillus velezensis* AGVL-005	94.92	CP024922.1	CU084_15155
*Bacillus velezensis* S141	94.92	AP018402.1	BVS141_RS20160
*Bacillus velezensis* G341	94.92	CP011686.1	ABH13_RS20940
*Bacillus velezensis* KS04AU	94.92	CP092750.1	MKF36_13400
*Bacillus velezensis* GMEKP1	94.92	CP076450.1	KOM03_18910
*Bacillus velezensis* BIOMA BV10	94.92	CP059318.1	HZT45_RS19200
*Bacillus velezensis* SRCM102742	94.92	CP028206.1	C7M20_RS13645
*Bacillus velezensis* SRCM102741	94.92	CP028205.1	C7M19_RS05115
*Bacillus velezensis* KD1	94.53	CP014990.2	A2I97_19425
*Bacillus velezensis* DMB05	94.92	CP083715.1	LAZ96_04145
*Bacillus velezensis* S4	94.42	CP050424.1	BVELS4_RS20280
*Bacillus velezensis* SRCM102752	94.42	CP028961.1	DBK22_RS19770
*Bacillus velezensis* SRCM102747	94.42	CP028211.1	C7M25_RS20685
*Bacillus velezensis* SRCM102746	94.42	CP028210.1	C7M24_RS19720
*Bacillus velezensis* SRCM102744	94.42	CP028208.1	C7M22_RS20530
*Bacillus velezensis* SRCM102743	94.42	CP028207.1	C7M21_RS19665
*Bacillus velezensis* UFLA258	94.42	CP039297.1	E4T61_13605
*Bacillus velezensis* 1B-23	94.42	CP033967.1	EG882_09440
*Bacillus velezensis* NKG-1	94.42	CP024203.1	CS376_RS21905
*Bacillus velezensis* GH1-13	94.42	CP019040.1	BVH55_14590
*Bacillus velezensis* LABIM44	94.42	CP079719.1	KXY09_14350
*Bacillus velezensis* Sam8H1	94.42	CP069391.1	JR311_14195
*Bacillus velezensis* AS43.3	94.42	CP003838.1	B938_RS20490
*Bacillus velezensis* KMU01	94.42	CP063768.1	IM712_RS10485
*Bacillus velezensis* AK-0	94.42	CP047119.1	GE573_RS19575
*Bacillus velezensis* Bac57	94.42	CP033054.1	D9777_RS16315
*Bacillus velezensis* W1	94.42	CP028375.1	C9888_RS15230
*Bacillus velezensis* BR-01	94.42	CP090150.1	LXH20_13635
*Bacillus velezensis* DMB07	94.42	CP083764.1	LAZ98_14795
*Bacillus velezensis* NZ4	94.42	CP076119.1	KM132_14025
*Bacillus velezensis* JK19	94.42	CP073781.1	KEM64_04660
*Bacillus velezensis* PEBA20	94.42	CP046145.1	GKO36_14775

“NA” = Not available.

**Table 3 microorganisms-10-01216-t003:** Real-time PCR and plating assay detection of strain ZF2 in different soil types.

Soil/Source (Province)	Copies/g (Colonies/g) 1 Day	Copies/g (Colonies/g) 7 Day	Copies/g (Colonies/g) 14 Day	Copies/g (Colonies/g) 21 Day	Copies/g (Colonies/g) 28 Day	Copies/g (Colonies/g) 35 Day
Black soil/Hei Long Jiang	1.48 × 10^7^	9.68 × 10^6^	5.92 × 10^6^	3.50 × 10^6^	1.35 × 10^6^	8.64 × 10^5^
	(3.59 × 10^8^)	(7.90 × 10^7^)	(2.12 × 10^7^)	(1.73 × 10^7^)	(3.79 × 10^7^)	(1.33 × 10^7^)
Red soil/Hai Nan	4.95 × 10^6^	3.92 × 10^6^	2.98 × 10^6^	2.42 × 10^6^	9.16 × 10^5^	6.27 × 10^5^
	(5.25 × 10^8^)	(1.23 × 10^6^)	(3.28 × 10^6^)	(6.22 × 10^6^)	(5.02 × 10^6^)	(5.25 × 10^6^)
Yellow brown soil/He Bei	8.08 × 10^7^	3.96 × 10^7^	2.31 × 10^7^	1.93 × 10^7^	1.22 × 10^7^	9.92 × 10^6^
	(4.99 × 10^8^)	(4.99 × 10^6^)	(1.59 × 10^5^)	(2.71 × 10^6^)	(1.05 × 10^6^)	(7.08 × 10^5^)
Brown soil/Jiang Su	4.12 × 10^7^	2.20 × 10^6^	1.65 × 10^6^	8.96 × 10^5^	5.77 × 10^5^	4.16 × 10^5^
	(2.79 × 10^8^)	(2.01 × 10^6^)	(3.54 × 10^5^)	(6.75 × 10^5^)	(5.75 × 10^5^)	(5.54 × 10^5^)
Desert soil/Ning Xia	3.40 × 10^8^	2.76 × 10^8^	1.63 × 10^8^	1.12 × 10^8^	1.01 × 10^8^	9.36 × 10^7^
	(3.64 × 10^8^)	(2.93 × 10^8^)	(2.44 × 10^8^)	(2.77 × 10^8^)	(1.59 × 10^8^)	(1.12 × 10^8^)

Values in parentheses represent the testing results of 10-fold dilution plating assays.

**Table 4 microorganisms-10-01216-t004:** The physicochemical properties of different types of soils.

Soil	pH	Organic Matter (g/kg)	Total N (g/kg)	Ammonium N (mg/kg)	Nitrate N (mg/kg)	Total P (mg/g)	Available P (mg/kg)	Total K (mg/g)	Available K (mg/kg)
Brown soil	8.28	29.65	1.97	6.35	26.81	1.49	151.42	16.30	450.5
Black soil	7.54	7.16	0.62	13.50	18.14	0.40	14.58	6.40	131.5
Red soil	8.07	18.00	1.44	21.81	31.40	1.04	75.15	17.10	368.5
Desert soil	8.40	3.27	0.42	24.02	43.75	0.51	2.93	14.75	90.5
Yellow brown soil	7.71	35.32	2.30	2.28	29.42	3.97	202.36	12.25	802.5

**Table 5 microorganisms-10-01216-t005:** The colonization abilities of strain ZF2 in different pH.

pH Value	Copies/g (Colonies/g) 1 Day	Copies/g (Colonies/g) 7 Day	Copies/g (Colonies/g) 14 Day	Copies/g (Colonies/g) 21 Day	Copies/g (Colonies/g) 28 Day	Copies/g (Colonies/g) 35 Day
4	1.23 × 10^6^	7.22 × 10^5^	4.06 × 10^5^	2.97 × 10^5^	1.86 × 10^5^	1.30 × 10^5^
	(1.35 × 10^7^)	(1.41 × 10^6^)	(1.31 × 10^6^)	(3.15 × 10^5^)	(4.27 × 10^5^)	(1.34 × 10^5^)
5	3.15 × 10^6^	1.36 × 10^6^	1.13 × 10^6^	4.53 × 10^5^	3.41 × 10^5^	2.34 × 10^5^
	(1.47 × 10^7^)	(3.42 × 10^6^)	(2.22 × 10^6^)	(1.73 × 10^6^)	(6.97 × 10^5^)	(4.33 × 10^5^)
6	3.70 × 10^7^	2.82 × 10^6^	2.02 × 10^6^	1.12 × 10^6^	6.68 × 10^5^	6.75 × 10^5^
	(7.83 × 10^7^)	(4.28 × 10^6^)	(6.75 × 10^6^)	(4.51 × 10^6^)	(9.05 × 10^6^)	(6.58 × 10^6^)
7	1.88 × 10^8^	4.42 × 10^7^	2.63 × 10^7^	2.52 × 10^7^	1.51 × 10^7^	8.54 × 10^6^
	(7.81 × 10^7^)	(1.57 × 10^7^)	(5.17 × 10^7^)	(1.24 × 10^7^)	(1.58 × 10^7^)	(1.02 × 10^7^)
8	6.70 × 10^7^	1.95 × 10^7^	1.45 × 10^7^	2.27 × 10^7^	1.43 × 10^7^	8.11 × 10^6^
	(7.11 × 10^7^)	(6.90 × 10^6^)	(2.16 × 10^7^)	(9.87 × 10^6^)	(1.25 × 10^7^)	(8.98 × 10^6^)
9	8.59 × 10^6^	2.81 × 10^6^	2.34 × 10^6^	2.22 × 10^6^	2.27 × 10^6^	1.55 × 10^6^
	(6.43 × 10^7^)	(6.51 × 10^6^)	(7.16 × 10^6^)	(4.21 × 10^6^)	(3.98 × 10^6^)	(4.56 × 10^6^)
10	1.57 × 10^6^	5.47 × 10^5^	2.82 × 10^5^	2.71 × 10^5^	1.97 × 10^6^	1.56 × 10^5^
	(3.17 × 10^7^)	(2.88 × 10^6^)	(4.61 × 10^6^)	(3.17 × 10^6^)	(2.15 × 10^6^)	(2.52 × 10^6^)

Values in parentheses represent the testing results of 10-fold dilution plating assays.

**Table 6 microorganisms-10-01216-t006:** The colonization abilities of strain ZF2 under different nutrient additions.

Nutrient	Copies/g (Colonies/g) 1 Day	Copies/g (Colonies/g) 7 Day	Copies/g (Colonies/g) 14 day	Copies/g (Colonies/g) 21 Day	Copies/g (Colonies/g) 28 Day	Copies/g (Colonies/g) 35 Day
CK	4.35 × 10^8^	1.10 × 10^6^	5.74 × 10^5^	1.18 × 10^5^	1.88 × 10^5^	8.42 × 10^4^
	(2.93 × 10^8^)	(6.00 × 10^5^)	(1.60 × 10^5^)	(7.05 × 10^4^)	(8.5 × 10^4^)	(7.22 × 10^4^)
Fructose	5.43 × 10^8^	3.39 × 10^6^	5.01 × 10^5^	6.85 × 10^4^	8.95 × 10^4^	1.56 × 10^5^
	(3.27 × 10^8^)	(8.27 × 10^6^)	(1.12 × 10^5^)	(2.12 × 10^4^)	(2.25 × 10^4^)	(1.28 × 10^5^)
Sucrose	4.37 × 10^8^	1.15 × 10^6^	6.34 × 10^5^	1.62 × 10^5^	1.97 × 10^5^	1.26 × 10^5^
	(2.63 × 10^8^)	(7.50 × 10^5^)	(1.52 × 10^5^)	(1.25 × 10^5^)	(1.25 × 10^5^)	(9.68 × 10^4^)
Maltose	3.37 × 10^8^	9.49 × 10^5^	2.37 × 10^5^	1.23 × 10^5^	1.90 × 10^5^	2.36 × 10^5^
	(9.21 × 10^7^)	(7.50 × 10^5^)	(1.05 × 10^5^)	(1.55 × 10^5^)	(1.55 × 10^5^)	(1.76 × 10^5^)
Molasses	6.56 × 10^8^	1.03 × 10^6^	1.73 × 10^5^	1.16 × 10^5^	1.10 × 10^5^	1.55 × 10^5^
	(2.03 × 10^8^)	(9.00 × 10^5^)	(8.15 × 10^4^)	(6.24 × 10^4^)	(6.16 × 10^4^)	(6.32 × 10^4^)
Dextrin	7.95 × 10^8^	2.75 × 10^6^	1.53 × 10^6^	1.42 × 10^5^	1.21 × 10^5^	1.30 × 10^5^
	(4.24 × 10^8^)	(3.55 × 10^6^)	(1.24 × 10^6^)	(8.48 × 10^4^)	(8.28 × 10^4^)	(7.60 × 10^4^)
Corn flour	9.44 × 10^8^	2.59 × 10^6^	6.21 × 10^5^	6.61 × 10^5^	3.665 × 10^5^	5.17 × 10^5^
	(8.65 × 10^8^)	(4.10 × 10^6^)	(3.90 × 10^5^)	(4.2 × 10^5^)	(3.73 × 10^5^)	(3.18 × 10^5^)
Yeast extract	5.61 × 10^8^	2.98 × 10^6^	1.35 × 10^6^	6.79 × 10^5^	5.11 × 10^5^	5.22 × 10^5^
	(2.41 × 10^8^)	(5.07 × 10^6^)	(1.04 × 10^6^)	(5.07 × 10^5^)	(2.53 × 10^5^)	(3.35 × 10^5^)
Peanut meal	7.69 × 10^8^	3.52 × 10^6^	1.14 × 10^6^	9.32 × 10^5^	6.51 × 10^5^	7.77 × 10^5^
	(7.91 × 10^8^)	(8.43 × 10^6^)	(1.30 × 10^6^)	(6.62 × 10^5^)	(4.06 × 10^5^)	(5.62 × 10^5^)
Peptone	8.44 × 10^8^	1.99 × 10^6^	1.04 × 10^6^	1.14 × 10^6^	1.51 × 10^6^	1.02 × 10^6^
	(6.97 × 10^8^)	(4.83 × 10^6^)	(1.37 × 10^6^)	(7.93 × 10^5^)	(1.07 × 10^6^)	(1.12 × 10^6^)
Soybean flour	6.51 × 10^8^	6.63 × 10^6^	1.11 × 10^6^	1.02 × 10^6^	5.65 × 10^5^	6.24 × 10^5^
	(6.5 × 10^8^)	(1.13 × 10^7^)	(1.74 × 10^6^)	(7.62 × 10^5^)	(7.67 × 10^5^)	(6.74 × 10^5^)
Soluble starch	5.27 × 10^8^	1.64 × 10^6^	4.59 × 10^5^	6.03 × 10^5^	1.76 × 10^5^	1.59 × 10^5^
	(3.51 × 10^8^)	(1.25 × 10^6^)	(3.93 × 10^5^)	(1.35 × 10^5^)	(9.52 × 10^4^)	(9.25 × 10^4^)
MgCl_2_	5.91 × 10^8^	7.19 × 10^5^	3.77 × 10^5^	1.45 × 10^5^	1.78 × 10^5^	1.32 × 10^5^
	(3.99 × 10^8^)	(5.50 × 10^5^)	(1.60 × 10^5^)	(7.06 × 10^4^)	(8.55 × 10^4^)	(7.20 × 10^4^)
CaCl_2_	5.59 × 10^8^	1.23 × 10^6^	3.37 × 10^5^	1.40 × 10^5^	1.07 × 10^5^	1.06 × 10^5^
	(3.78 × 10^8^)	(3.87 × 10^6^)	(6.00 × 10^5^)	(5.05 × 10^4^)	(6.25 × 10^4^)	(5.62 × 10^4^)
FeSO_4_	3.35 × 10^8^	8.39 × 10^5^	3.80 × 10^5^	1.23 × 10^5^	1.25 × 10^5^	1.19 × 10^5^
	(7.75 × 10^7^)	(1.05 × 10^6^)	(3.77 × 10^5^)	(1.16 × 10^5^)	(6.05 × 10^4^)	(5.04 × 10^4^)
KCl	4.24 × 10^8^	5.96 × 10^5^	5.35 × 10^5^	1.18 × 10^5^	1.46 × 10^5^	1.23 × 10^5^
	(1.02 × 10^8^)	(8.00 × 10^5^)	(7.97 × 10^5^)	(2.21 × 10^4^)	(5.28 × 10^4^)	(5.52 × 10^4^)
NaCl	5.31 × 10^8^	6.79 × 10^6^	3.19 × 10^5^	1.46 × 10^5^	1.75 × 10^5^	1.38 × 10^5^
	(1.38 × 10^8^)	(9.05 × 10^5^)	(1.20 × 10^5^)	(6.53 × 10^4^)	(6.20 × 10^4^)	(6.02 × 10^4^)

Values in parentheses represent the testing results of 10-fold dilution plating assays.

## Data Availability

The sequence of the D3N19_RS13500 gene of *B. velezensis* ZF2 can be obtained in the complete genome sequence of strain ZF2. The complete genome sequence of *Bacillus velezensis* ZF2 has been deposited in NCBI under the GenBank accession number CP032154.1.

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
