# Peer review of "Development of a Real-Time Quantitative PCR Assay for the Specific Detection of Bacillus velezensis and Its Application in the Study of Colonization Ability"

_microorganisms, 2022, doi:10.3390/microorganisms10061216_

Round 1
Reviewer 1 Report
Comments to the Author
General comments: The manuscript by Xu et al., (submitted) describes the development of a real-time quantitative PCR assay for quantitative detection of Bacillus velezensis and its colonization ability. This is an interesting study which establishes a rapid method for detection of B. velezensis and its application. In general, the justification of this work is sound, results are clear and the manuscript has been well written. However, for this manuscript to be accepted, kindly address specific comments below.
Specific comments
Lines 44-45: Do you have a reference for this claim?
Line 56: Rephrase “established a new rapid and specific detection….”
Lines 26-60: In the entire introduction section, I am missing references that cite similar work which has been done aimed at detecting B. amyloliquefaciens (see Johansson et al., 2014) and B. subtilis (see Xie et al., 2019). The inclusion of this information in this section will give a broader baseline view for your work and further make it contextual in the field of Bacillus research.
Line 191: I propose that you move Table 2 to the supplementary section. It is a bit distracting in the main text.
Lines 301-371: I found it interesting that your gene target was not amplified in strains FZB42T and SQR9. Do you have any idea if this challenge was encountered by other authors in their attempt to design specific quantitative detection methods for Bacillus species? I know this to be the case with Pseudomonas species. This may suggest that this target gene, may be conserved according to taxonomic placement and may require the design of a separate primer set specific for amplification of these exclusive strains. Perhaps you should mention this in your discussion too.

Author Response
Point 1: Lines 44-45: Do you have a reference for this claim?
Response 1: Thank you for your suggestion. We have improved the claim to “However, taking plant-beneficial microorganisms from lab to agricultural application is a great challenge, as exogenous inoculum is usually eliminated in soils due to competition from indigenous microbes or complicated soil environments. Research on these biocontrol agents indicated that the biocontrol efficacy of the microbes to control plant diseases was related to their ability to survive and maintain an abundant population in rhizosphere soil. Therefore, it is important to detect the viability of inoculum in soil.” on line 51-57 in revised manuscript. Meanwhile, the relevant reference was supplemented in the test and the reference list of revised manuscript.
Point 2: Line 56: Rephrase “established a new rapid and specific detection….”.
Response 2: Thanks for your opinion, the phrase “established a new rapid, and specific detection system” has been changed to “A new rapid and specific detection system was established” on line 70 in revised manuscript.
Point 3: Lines 26-60: In the entire introduction section, I am missing references that cite similar work which has been done aimed at detecting B. amyloliquefaciens (see Johansson et al., 2014) and B. subtilis (see Xie et al., 2019). The inclusion of this information in this section will give a broader baseline view for your work and further make it contextual in the field of Bacillus research.
Response 3: Thanks for your opinion. Considering the reviewer’s suggestion, we have supplemented the relevant materials on the detection of B. amyloliquefaciens and B. subtilis “Previous studies showed that the specific genes trpE, yecA and tetB were used to specifically detect three closely related B. amyloliquefaciens strains UCMB5033, UCMB5036 and UCMB5113, respectively. The primer pair designed based on the unique gene ukfpg of B. anyloliquefaciens TF28 was able to distinguish strain TF28 from other Bacillus strains. In addition, a colorimetric assay easily detected B. subtilis from Escherichia coli and Staphylococcus aureus. However, very few studies have been used to specifically detect B. velezensis strains or distinguish B. velezensis from B. amyloliquefaciens” on line 33-40 in revised manuscript to improve the introduction section. And three corresponding references were added to the revised manuscript.
Point 4: Line 191: I propose that you move Table 2 to the supplementary section. It is a bit distracting in the main text.
Response 4: We agree with your opinion. The Table 2 has been changed to Table S1 and moved to supplementary section in the revised manuscript.
Point 5: Lines 301-371: I found it interesting that your gene target was not amplified in strains FZB42T and SQR9. Do you have any idea if this challenge was encountered by other authors in their attempt to design specific quantitative detection methods for Bacillus species? I know this to be the case with Pseudomonas species. This may suggest that this target gene, may be conserved according to taxonomic placement and may require the design of a separate primer set specific for amplification of these exclusive strains. Perhaps you should mention this in your discussion too.
Response 5: Thank you very much four your suggestion. At present, most studies of specific quantitative detection on Bacillus focused on a certain strain, very few studies focused on the specific detection of species, so we have not found any other studies which had the similar challenge in quantitative detection methods for Bacillus species. In the study, we compared the gene sequences of many B. velezensis strains and B. amyloliquefaciens, B. subtilis strains, then screened out the specific primer. In the specificity verification test, we chose many B. velezensis strains which have been well studied or have been confirmed the taxonmic position clearly. Interestingly, there were no signals in PCR test when the genomic DNA of strain FZB42 and SQR9 used as template. However, the target gene in our study could be retrieved in most B. velezensis strains according to the NCBI database. We thought it was reasonable for the occasionality in scientific research. Considering the valuable opinion of reviewer, we have supplemented further discussion on this interesting phenomenon on line 351-356 in revised manuscript.
Reviewer 2 Report
The manuscript is dedicated to the study of Bacillus velezensis bacteria, more precisely to the development of the PCR-based detection system. The quality of the manuscript is high and the results are interesting, though there are some remarks:
1. The manuscripts consists of two parts: development of PCR based system and studying colonization properties. It would be better to split the manuscript into two independent articles or shift the accent from the second part.
2. In the manuscript the authors mentioned that not all B. velezensis gave the signal in PCR test. It requires deeper study.
3. Do you have the genome sequences FZB42 and SQR9 strains? Do they contain the target region?
4. You mentioned, that target sequence for PCR is presented in the most part of B. velezensis genome. What is "in the most part"?
5. Have you check that other genomes of Bacillus genus does not contain this sequence, apart from the test ones?
6. The part with colonization requires more discussion and explaining the mechanisms.
Author Response
Point 1: The manuscripts consists of two parts: development of PCR based system and studying colonization properties. It would be better to split the manuscript into two independent articles or shift the accent from the second part.
Response 1: Thank you for your kind advice. Considering the reviewer’s suggestion, the article was divided into two main parts “Establishment of specific detection system” and “Detection of the colonization ability of B. velezensis ZF2” in the revised manuscript. Meanwhile, we supplemented subsubsection in the revised manuscript to improve the structure of the article more clear.
Point 2: In the manuscript the authors mentioned that not all B. velezensis gave the signal in PCR test. It requires deeper study.
Response 2: Yes, we agree with your opinion and thanks four your suggestion. In the study, we compared the gene sequences of many B. velezensis strains and B. amyloliquefaciens, B. subtilis strains, then screened out the specific primer. In the specificity verification test, we chose many B. velezensis strains which have been well studied or have been confirmed the taxonmic position clearly. And there were no signal in PCR test when the genomic DNA of strain FZB42 and SQR9 used as template. However, the target gene in our study could be retrieved in most B. velezensis strains according to the NCBI database. We thought it was reasonable for the occasionality in scientific research, and discussed the finding in the paper. We inferred that this may be related to the taxonmic position of strains FZB42 and SQR9, as the fact that B. velezensis FZB42 and SQR9 were originally classified as B. amyloliquefaciens. Of course, it would be meaningful to reveal this interesting phenomenon through deeper study.
Point 3: Do you have the genome sequences FZB42 and SQR9 strains? Do they contain the target region?
Response 3: Thank you for your suggestion. Yes, we obtained the genome sequences of FZB42 (GenBank: CP000560.2) and SQR9 (GenBank: CP006890.1) from NCBI databsed and searched the target region. And the genome sequences of two strains did not contain the target region.
Point 4: You mentioned, that target sequence for PCR is presented in the most part of B. velezensis genome. What is "in the most part"?
Response 4: Thanks for your suggestion and we are sorry for the confusing sentence. The sentence should be “the gene could be retrieved in most B. velezensis strains” on line 354 not “in the most part of B. velezensis”. In the study, we searched the target sequence of amplified fragement by NCBI dtabase through nucleotide BLAST, and most B. velezensis strains were retrieved (Table 2). We mean that the target sequence are present in most B. velezensis strains.
Point 5: Have you check that other genomes of Bacillus genus does not contain this sequence, apart from the test ones?
Response 5: Thank you for your opinion. Yes, we checked many genomes of B. amyloliquefaciens, B. subtilis strains, and they did not contain the target gene.
Point 6: The part with colonization requires more discussion and explaining the mechanisms.
Response 6: Thank you for your suggestion. We have supplemented further discussion on colonization and the mechanisms. The supplementary details were on line 373-378 and line 385-388 in revised manuscript.
Reviewer 3 Report
This is a straightforward study.
The experiments are appropriate and the overall conclusions seem correct.
Furthermore the overall topic is of interest given the biotechnological importance of these microorganisms.
I have no major nor minor comments for the authors consideration.
Author Response
Point: This is a straightforward study.
The experiments are appropriate and the overall conclusions seem correct.
Furthermore the overall topic is of interest given the biotechnological importance of these microorganisms.
I have no major nor minor comments for the authors consideration.
Response: Thank you very much for your comments and suggestions.